# Pruning Filter in Filter

**Fanxu Meng**[1,2*]**, Hao Cheng**[2*]**,**
**Ke Li**[2]**, Huixiang Luo**[2]**, Xiaowei Guo**[2]**, Guangming Lu**[1†]**, Xing Sun**[2†]
[1] Harbin Institute of Technology, Shenzhen, China
[2] Tencent Youtu Lab, Shanghai, China
`18S151514@stu.hit.edu.cn, luguangm@hit.edu.cn`
`{louischeng, tristanli,huixiangluo,scorpioguo,winfredsun}@tencent.com`

## Abstract

Pruning has become a very powerful and effective technique to compress and accelerate modern neural networks. Existing pruning methods can be grouped into two categories: filter pruning (FP) and weight pruning (WP). FP wins at hardware compatibility but loses at the compression ratio compared with WP. To converge the strength of both methods, we propose to prune the filter in the filter. Specifically, we treat a filter $F \in \mathbb{R}^{C \times K \times K}$ as $K \times K$ stripes, *i.e.*, $1 \times 1$ filters $\in \mathbb{R}^C$, then by pruning the stripes instead of the whole filter, we can achieve finer granularity than traditional FP while being hardware friendly. We term our method as SWP (*Stripe-Wise Pruning*). SWP is implemented by introducing a novel learnable matrix called Filter Skeleton, whose values reflect the shape of each filter. As some recent work has shown that the pruned architecture is more crucial than the inherited important weights, we argue that the architecture of a single filter, *i.e.*, the shape, also matters. Through extensive experiments, we demonstrate that SWP is more effective compared to the previous FP-based methods and achieves the state-of-art pruning ratio on CIFAR-10 and ImageNet datasets without obvious accuracy drop. Code is available at this url.

## 1 Introduction

Deep Neural Networks (DNNs) have achieved remarkable progress in many areas including speech recognition [1], computer vision [2, 3], natural language processing [4], *etc*. However, model deployment is sometimes costly due to the large number of parameters in DNNs. To relieve such a problem, numerous approaches have been proposed to compress DNNs and reduce the amount of computation. These methods can be classified into two main categories: weight pruning (WP) and filter (channel) pruning (FP).

WP is a fine-grained pruning method that prunes the individual weights, *e.g.*, whose value is nearly 0, inside the network [5, 6], resulting in a sparse network without sacrificing prediction performance. However, since the positions of non-zero weights are irregular and random, we need an extra record of the weight position, and the sparse network pruned by WP can not be presented in a structured fashion like FP due to the randomness inside the network, making WP unable to achieve acceleration on general-purpose processors. By contrast, FP-based methods [7, 8, 9] prune filters or channels within the convolution layers, thus the pruned network is still well organized in a structure fashion and can easily achieve acceleration in general processors. A standard filter pruning pipeline is as follows: 1) Train a larger model until convergence. 2) Prune the filters according to some criterions 3) Fine-tune the pruned network. [10] observes that training the pruned model with random initialization

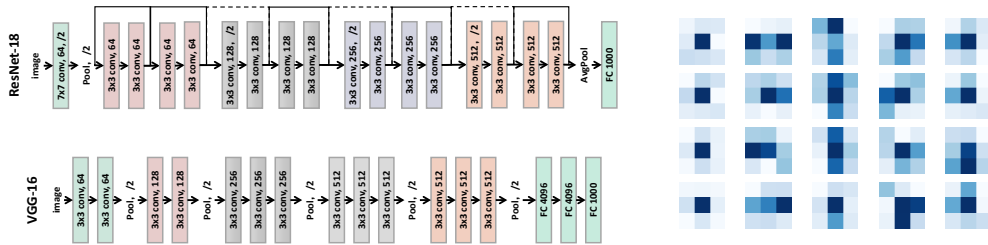

Figure 1: The left figure shows two network structures. The right figure visualizes the average $l_1$ norm of the filters along the channel dimension in a learned VGG16.

can also achieve high performance. Thus it is the network architecture, rather than trained weights that matters. In this paper, we suggest that *not only the architecture of the network but the architecture of the filter itself is also important*. [11, 12] also draw similar arguments that the filter with a larger kernel size may lead to better performance. However, the computation cost is expensive. Thus for a given input feature map, [11, 12] uses filters with different kernel sizes (*e.g.,* $1 \times 1$, $3 \times 3$, *and* $5 \times 5$) to perform convolution and concatenate all the output feature map. But the kernel size of each filter is manually set. It needs professional experience and knowledge to design an efficient network structure. We wonder *what if we can learn the optimal kernel size of each filter by pruning*. Our intuition is illustrated in Figure 1. We know that the structure of deep nets matters for learning tasks. For example, the residual net is easier to optimize and exhibits better performance than VGG. However, we find that there is another structure hidden inside the network, which we call **'the shape of the filters'**. From Figure 1, not all the stripes in a filter contribute equally [13]. Some stripes have a very low $l_1$ norm indicating that such stripes can be removed from the network. The optimal shape of the filter is the filter with minimal stripes that maintains the function of the filter. To capture the 'filter shape' alongside the filter weights, we propose 'Filter Skeleton (FS)' to learn this 'shape' property and use FS to guide efficient pruning (i.e. learn optimal shape) (See Section 3). Compared to the traditional FP-based pruning, this pruning paradigm achieves finer granularity since we operate with stripes rather than the whole filter.

Similarly, group-wise pruning, introduced in [14, 15, 16] also achieves finer granularity than filter/channel pruning, which removes the weights located in the same position among all the filters in a certain layer. However, group-wise pruning breaks the independent assumption on the filters. For example, the invalid positions of weights in each filter may be different. By regularizing the network using group-wise pruning, the network may lose representation ability under a large pruning ratio (see Section 4.2). In this paper, we also offer a comparison to group-wise pruning in the experiment. In contrast, SWP keeps each filter independent with each other which does not break the independent assumption among the filters. Throughout the experiments, SWP achieves a higher pruning ratio compared to the filter-wise, channel-wise, and group-wise pruning methods. We summarize our main contributions below:

- We propose a new pruning paradigm called SWP. SWP achieves a finer granular than traditional filter pruning and the pruned network can still be inferred efficiently.
- We introduce Filter Skeleton (FS) to efficiently learn the shape of each filter and deeply analyze the working mechanism of FS. Using FS, we achieve the state-of-art pruning ratio on CIFAR-10 and ImageNet datasets without obvious accuracy drop.

## 2   Related Work

**Weight pruning:** Weight pruning (WP) dates back to optimal brain damage and optimal brain surgeon [17, 18], which prune weights based on the Hessian of the loss function. [5] prunes the network weights based on the $l_1$ norm criterion and retrain the network to restore the performance and this technique can be incorporated into the deep compression pipeline through pruning, quantization, and Huffman coding [6]. [19] reduces the network complexity by making on-the-fly connection

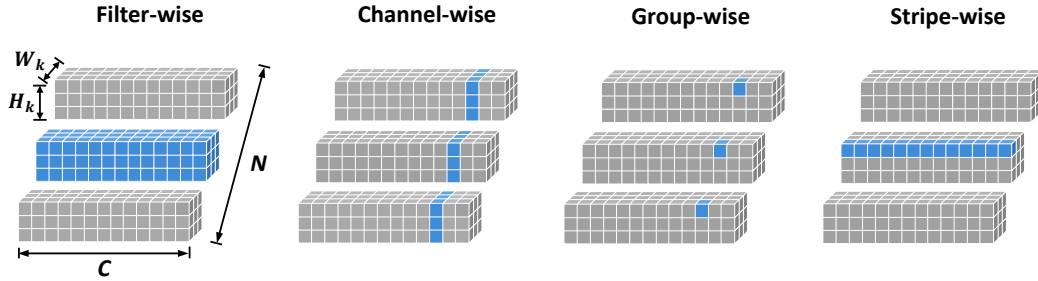

Figure 2: The visualization of different types of pruning.

pruning, which incorporates connection splicing into the whole process to avoid incorrect pruning and make it as continual network maintenance. [20] removes connections at each DNN layer by solving a convex optimization program. This program seeks a sparse set of weights at each layer that keeps the layer inputs and outputs consistent with the originally trained model. [21] proposes a frequency-domain dynamic pruning scheme to exploit the spatial correlations on CNN. The frequency-domain coefficients are pruned dynamically in each iteration and different frequency bands are pruned discriminatively, given their different importance on accuracy. [22] divides each stripe into multiple groups, and prune weights in each group. However, one drawback of these unstructured pruning methods is that the resulting weight matrices are sparse, which cannot lead to compression and speedup without dedicated hardware/libraries [23].

**Filter/Channel Pruning:** Filter/Channel Pruning (FP) prunes at the level of filter, channel, or even layer. Since the original convolution structure is still preserved, no dedicated hardware/libraries are required to realize the benefits. Similar to weight pruning [5], [7] also adopts $l_1$ norm criterion that prunes unimportant filters. Instead of pruning filters, [8] proposed to prune channels through LASSO regression-based channel selection and least square reconstruction. [9] optimizes the scaling factor $\gamma$ in the BN layer as a channel selection indicator to decide which channel is unimportant and can be removed. [24] introduces ThiNet that formally establish filter pruning as an optimization problem, and reveal that we need to prune filters based on statistics information computed from its next layer, not the current layer. Similarly, [25] optimizes the reconstruction error of the final response layer and propagates an 'importance score' for each channel. [26] first proposes that utilize AutoML for Model Compression which leverages reinforcement learning to provide the model compression policy. [27] proposes an effective structured pruning approach that jointly prunes filters as well as other structures in an end-to-end manner. Specifically, the authors introduce a soft mask to scale the output of these structures by defining a new objective function with sparsity regularization to align the output of the baseline and network with this mask. [28] introduces a budgeted regularized pruning framework for deep CNNs that naturally fit into traditional neural network training. The framework consists of a learnable masking layer, a novel budget-aware objective function, and the use of knowledge distillation. [29] proposes a global filter pruning algorithm called Gate Decorator, which transforms a vanilla CNN module by multiplying its output by the channel-wise scaling factors, i.e. gate, and achieves state-of-art results on CIFAR dataset. [30, 10] deeply analyze how initialization affects pruning through extensive experimental results.

**Group-wise Pruning:** [14, 15] introduces group-wise pruning, that learns a structured sparsity in neural networks using group lasso regularization. The group-wise pruning can still be efficiently processed using 'im2col' implementation as filter-wise and channel-wise pruning. [31] further explores a complete range of pruning granularity and evaluate how it affects the prediction accuracy. [16] improves the group-wise pruning by proposing a dynamic regularization method. However, group-wise pruning removes the weights located in the same position among all the filters in a certain layer. Since invalid positions of each filter may be different, group-wise pruning may cause the network to lose valid information. In a contrast, our approach keeps each filter independent with each other, thus can lead to a more efficient network structure. Different types of pruning are illustrated in Figure 2.

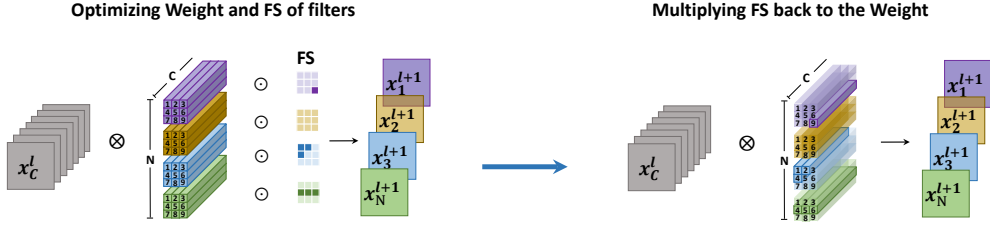

Figure 3: Traing and inference with Filter Skeleton (FS).

Table 1: Test accuracy of each network that only learns the 'shape' of the filters.

| Dataset | Backbone | Test accuracy |
|---------|----------|---------------|
|          | VGG16 | 79.83 |
| CIFAR-10 | ResNet56 | 83.82 |
|          | MobileNetV2 | 83.52 |

**Mask in Pruning:** Using a (soft) mask to represent the importance of component in network has been thoroughly studied in the work of pruning [32, 33, 34, 35, 27, 36, 37, 9, 8]. However, most work design the masks in terms of the filter or channels, few works pay attention to stripes. Also, Filter Skeleton (FS) is not just a mask, we consider each filter has two properties: weight and shape. FS is to learn the 'shape' property. From Section 3.1 in the paper, the network still has a good performance by only learning the 'shape' of the filters, keeping the filter weight randomly initialized.

## 3 The proposed Method

### 3.1 Filter Skeleton (FS)

FS is introduced to learn another important property of filters alongside their weight: *shape*, which is a matrix related to the stripes of the filter. Suppose the $l$-th convolutional layer's weight $W^l$ is of size $\mathbb{R}^{N \times C \times K \times K}$, where $N$ is the number of the filters, $C$ is the channel dimension and $K$ is the kernel size. Then the size of FS in this layer is $\mathbb{R}^{N \times K \times K}$. *I.e.,* each value in FS corresponds to a strip in the filter. FS in each layer is firstly initialized with all-one matrix. During training, We multiply the filters' weights with FS. Mathematically, the loss is represented by:

$$L = \sum_{(x,y)} loss(f(x, W \odot I), y) \tag{1}$$

, where $I$ represents the FS, $\odot$ denotes dot product. With $I$, the forward process is:

$$X^{l+1}_{n,h,w} = \sum_{c}^{C} \sum_{i}^{K} \sum_{j}^{K} I^l_{n,i,j} \times W^l_{n,c,i,j} \times X^l_{n,h+i-\frac{K+1}{2},w+j-\frac{K+1}{2}} \tag{2}$$

. The gradient with regard to $W$ and $I$ is:

$$grad(W^l_{n,c,i,j}) = I^l_{n,i,j} \times \sum_{h}^{M_H} \sum_{w}^{M_W} \frac{\partial L}{\partial X^{l+1}_{n,h,w}} \times X^l_{c,h+i-\frac{K+1}{2},w+j-\frac{K+1}{2}} \tag{3}$$

$$grad(I^l_{n,i,j}) = \sum_{c}^{C}(W^l_{n,c,i,j} \times \sum_{h}^{M_H} \sum_{w}^{M_W} \frac{\partial L}{\partial X^{l+1}_{n,h,w}} \times X^l_{c,h+i-\frac{K+1}{2},w+j-\frac{K+1}{2}}) \tag{4}$$

, where $M_H$, $M_W$ represent the height and width of the feature map respectively. $X^l_{c,p,q} = 0$ when $p < 1$ or $p > M_H$ or $q < 1$ or $q > M_W$ (this corresponds to the padding and shifting[38, 39] procedures).

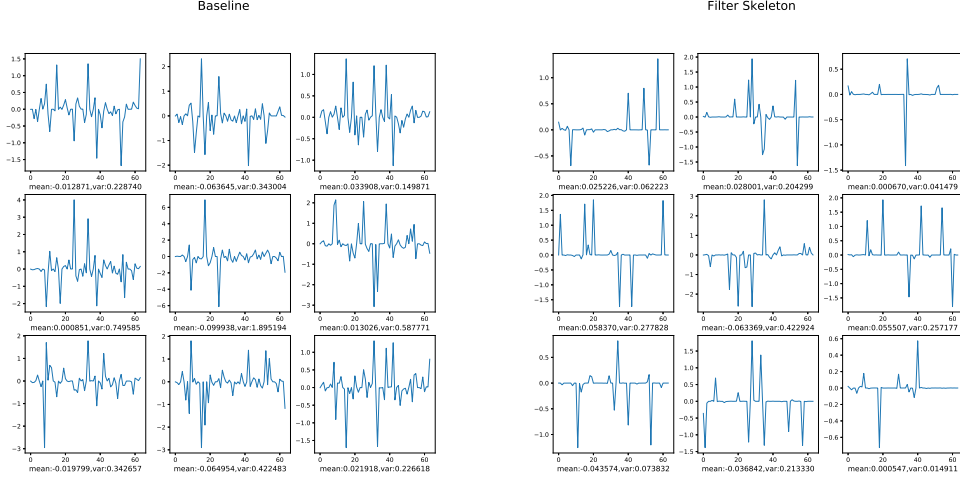

Figure 4: This left and the right figure shows the distribution of the weights of baseline and FS on the first convolution layer, respectively. In this layer, each filter has 9 strips. Each mini-figure shows the summation (y-axis) of the stripes located in the same position of all the filters (x-axis). The mean and std are also reported.

From (1), the filter weights and FS are jointly optimized during training. After training, we merge $I$ onto the filter weights $W$(*I.e.,* $W \leftarrow W \odot I$), and only use $W$ during evaluating. Thus no additional cost is brought to the network when applying inference. The whole process is illustrated in Figure 3. To further show the importance of the 'shape' property, we conduct an experiment where the filters' weights are fixed, only FS can be optimized during training. The results are shown on Table 1. It can be seen that without updating the filters' weights, the network still gets decent results. We also find that with Filter Skeleton, the weights of the network become more stable. Figure 4 displays the distribution of the weights of the baseline network and network trained by Filter Skeleton (FS). It can be seen that the weights trained by FS are sparse and smooth, which have a low variance to input images, leading to stable outputs. Thus the network is robust to the variation of the input data or features.

## 3.2 Stripe-wise pruning with FS

From Figure 1, not all the stripes contribute equally in the network. To build a compact and highly pruned network, the Filter Skeleton (FS) needs to be sparse. *I.e.,* when some values in FS is close to 0, the corresponding stripes can be pruned. Therefore, when training the network with FS, we impose regularization on FS to make it sparse:

$$L = \sum_{(x,y)} loss(f(x, W \odot I), y) + \alpha g(I) \tag{5}$$

, where $\alpha$ controls the magnitude of regularization, $g(I)$ indicates $l_1$ norm penalty on $I$, which is commonly used in many pruning approaches [7, 8, 9]. Specifically, $g(I)$ is written as:

$$g(I) = \sum_{l=1}^{L} g(I^l) = \sum_{l=1}^{L} (\sum_{n=1}^{N} \sum_{i=1}^{K} \sum_{j=1}^{K} |I_{n,i,j}^l|). \tag{6}$$

From (5), FS implicitly learns the optimal shape of each filter. In Section 4.4, we visualize the shape of filters to further show this phenomenon. To lead efficient pruning, we set a threshold $\delta$, the stripes whose corresponding values in FS are smaller than $\delta$ will not be updated during training and can be pruned afterwards. It is worth noticing that when performing inference on the pruned network, we can not directly use the filter as a whole to perform convolution on the input feature map since the filter is broken. Instead, we need to use each stripe independently to perform convolution and sum the feature map produced by each stripe, as shown in Figure 5. Mathematically, the convolution process

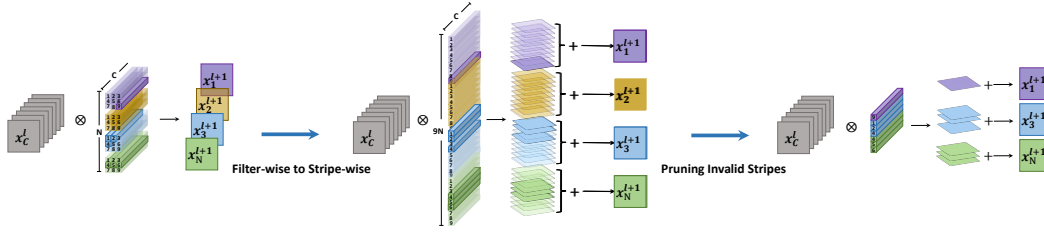

Figure 5: Pruning process in SWP.

in SWP is written as:

$$X_{n,h,w}^{l+1} = \sum_{c}^{C} \sum_{i}^{K} \sum_{j}^{K} W_{n,c,i,j}^{l} \times X_{n,h+i-\frac{K+1}{2},w+j-\frac{K+1}{2}}^{l} \quad standard\ convolution$$

$$= \sum_{i}^{K} \sum_{j}^{K} (\sum_{c}^{C} W_{n,c,i,j}^{l} \times X_{n,h+i-\frac{K+1}{2},w+j-\frac{K+1}{2}}^{l}) \quad stripe\ wise\ convolution$$

(7)

, where $X_{n,h,w}^{l+1}$ is one point of the feature map in the $l + 1$-th layer. From (7), SWP only modifies the calculation order in the conventional convolution process, thus no additional operations (Flops) are added to the network. It is worth noting that, since each stripe has its own position in the filter. SWP needs to record the indexes of all the stripes. However, it costs little compared to the whole network parameters. Suppose the $l$-th convolutional layer's weight $W^l$ is of size $\mathbb{R}^{N \times C \times K \times K}$. For SWP, we need to record $N \times K \times K$ indexes. Compared to the individual weight pruning which records $N \times C \times K \times K$ indexes, we reduce the weight pruning's indexes by $C$ times. Also, we do not need to record the indexes of the filter if all the stripes in such filter are removed from the network, and SWP degenerates to conventional filter-wise pruning. For a fair comparison with traditional FP-based methods, we add the number of indexes when calculating the number of network parameters.

There is two advantage of SWP compared to the traditional FP-based pruning:

- Suppose the kernel size is $K \times K$, then SWP achieves $K^2 \times$ finer granularity than traditional FP-based pruning, which leads to a higher pruning ratio.

- For certain datasets, *e.g.,* CIFAR-10, the network pruned by SWP keeps high performance even without a fine-tuning process. This separates SWP from many other FP-based pruning methods that require multiple fine-tuning procedures. The reason is that FS learns an optimal shape for each filter. By pruning unimportant stripes, the filter does not lose much useful information. In contrast, FP pruning directly removes filters which may damage the information learned by the network.

## 4   Experiments

This section arranges as follows: In Section 4.1, we introduce the implementation details in the paper; in Section 4.2, we compare SWP with group-wise pruning; in Section 4.3, we show SWP achieves state-of-art pruning ratio on CIFAR-10 and ImageNet datasets compared to filter-wise, channel-wise or shape-wise pruning; in Section 4.4, we visualize the pruned filters; in Section 4.5, we perform ablation studies to study how hyper-parameters influence SWP.

### 4.1   Implementation Details

**Datasets and Models:** CIFAR-10 [40] and ImageNet [41] are two popular datastes and are adopted in our experiments. CIFAR-10 dataset contains 50K training images and 10K test images for 10 classes. ImageNet contains 1.28 million training images and 50K test images for 1000 classes. On CIFAR-10, we evaluated our method on two popular network structures: VGG16 [42], ResNet56 [43]. On ImageNet dataset, we adopt ResNet18.

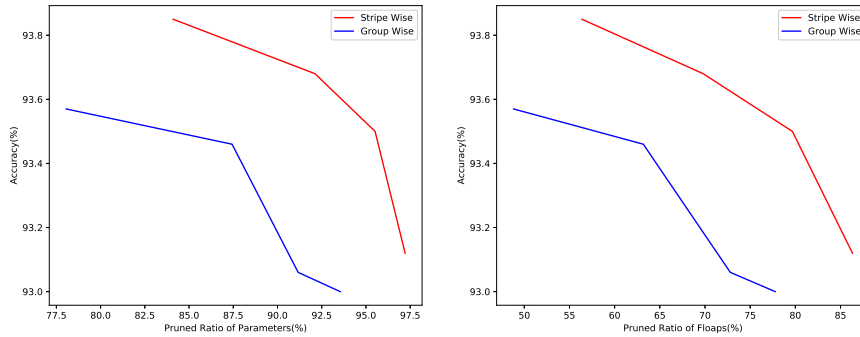

Figure 6: Comparing SWP with group-wise pruning on CIFAR-10. The backbone is VGG16.

**Baseline Setting:** Our baseline setting is consistent with [9]. For CIFAR-10, the model was trained for 160 epochs with a batch size of 64. The initial learning rate is set to 0.1 and divide it by 10 at the epoch 80 and 120. The simple data augmentation (random crop and random horizontal flip) is used for training images. For ImageNet, we follow the official PyTorch implementation [1] that train the model for 90 epochs with a batch size of 256. The initial learning rate is set to 0.1 and divide it by 10 every 30 epochs. Images are resized to $256 \times 256$, then randomly crop a $224 \times 224$ area from the original image for training. The testing is on the center crop of $224 \times 224$ pixels.

**SWP setting:** The basic hyper-parameters setting is consistent with the baseline. $\alpha$ is set to 1e-5 in (5) and the threshold $\delta$ is set to 0.05. For CIFAR-10, we do not fine-tune the network after stripe selection. For ImageNet, we perform a one-time fine-tuning after pruning.

### 4.2 Group-wise pruning *vs* stripe-wise pruning

Since group-wise pruning can also be implemented via Skeleton, we perform group-wise pruning and SWP both based on the Skeleton. Figure 6 shows the results. We can see under the same number of parameters or Flops, SWP achieves a higher performance compared to group-wise pruning. We also find that in group-wise pruning, $layer2.7.conv1$ and $layer2.7.conv2$ will be identified as invalid (*i.e.,* all the weights in such layer will be pruned by the algorithm) when the pruning ratio reaches 76.64%. However, this phenomenon does not appear at stripe-wise pruning even with an 87.36% pruning ratio, which further verifies our hypothesis that group-wise pruning breaks the independent assumption on the filters and may easily lose representation ability. In contrast, SWP keeps each filter independent of each other, thus can achieve a higher pruning ratio.

### 4.3 Comparing SWP with state-of-art methods

We compare SWP with recent state-of-arts pruning methods. Table 2 and Table 3 lists the comparison on CIFAR-10 and ImageNet, respectively. In Table 2, IR [16] is group-wise pruning method, the others except SWP are filter-wise or channel-wise methods. We can see GBN [29] even outperforms the shape-wise pruning method. From our analysis, group-wise pruning regularizes the network's weights in the same positions among all the filters, which may cause the network to lose useful information. Thus group-wise pruning may not be the best choice. However, SWP outperforms other methods by a large margin. For example, when pruning VGG16, SWP can reduce the number of parameters by 92.66% and the number of Flops by 71.16% without losing network performance. On ImageNet, SWP could also achieve better performance than recent benchmark approaches. For example, SWP can reduce the FLOPs by 54.58% without an obvious accuracy drop. We want to emphasize that even though SWP brings indexes of strips, the cost is little. When performing calculation on the number of parameters, We have added these indexes in the calculation on Table 2 and Table 3. The pruning ratio of SWP is still significant and achieves state-of-art results.

Table 2: Comparing SWP with state-of-arts FP-based methods on CIFAR-10 dataset. The baseline accuracy of ResNet56 is 93.1% [29], while VGG16's baseline accuracy is 93.25% [7].

| Backbone | Metrics | Params(%)↓ | FLOPS(%)↓ | Accuracy(%)↓ |
|---|---|---|---|---|
| VGG16 | L1[7] (ICLR 2017) | 64 | 34.3 | -0.15 |
| | ThiNet[24] (ICCV 2017) | 63.95 | 64.02 | 2.49 |
| | SSS[44] (ECCV 2018) | 73.8 | 41.6 | 0.23 |
| | SFP[45] (IJCAI 2018) | 63.95 | 63.91 | 1.17 |
| | GAL[27] (CVPR 2019) | 77.6 | 39.6 | 1.22 |
| | Hinge[46] (CVPR 2020) | 80.05 | 39.07 | -0.34 |
| | HRank[47] (CVPR 2020) | 82.9 | 53.5 | -0.18 |
| | Ours | **92.66** | **71.16** | **-0.4** |
| ResNet56 | L1[7] (ICLR 2017) | 13.7 | 27.6 | -0.02 |
| | CP[8] (ICCV 2017) | - | 50 | 1.00 |
| | NISP[25] (CVPR 2018) | 42.6 | 43.6 | 0.03 |
| | DCP[48] (NeurIPS 2018) | 70.3 | 47.1 | -0.01 |
| | IR[16] (IJCNN 2019) | - | 67.7 | 0.4 |
| | C-SGD[49] (CVPR 2019) | - | 60.8 | -0.23 |
| | GBN [29] (NeurIPS 2019) | 66.7 | 70.3 | 0.03 |
| | HRank[47] (CVPR 2020) | 68.1 | 74.1 | 2.38 |
| | Ours | **77.7** | **75.6** | **0.12** |

Table 3: Comparing SWP with state-of-arts pruning methods on ImageNet dataset. All the methods use ResNet18 as the backbone and the baseline top-1 and top-5 accuracy are 69.76% and 89.08%, respectively.

| Backbone | Metrics | FLOPS(%)↓ | Top-1(%)↓ | Top-5(%)↓ |
|---|---|---|---|---|
| ResNet18 | LCCL[50] (CVPR 2017) | 35.57 | 3.43 | 2.14 |
| | SFP[45] (IJCAI 2018) | 42.72 | 2.66 | 1.3 |
| | FPGM[51] (CVPR 2019) | 42.72 | 1.35 | 0.6 |
| | TAS[52] (NeurIPS 2019) | 43.47 | 0.61 | -0.11 |
| | DMCP[53] (CVPR 2020) | 42.81 | 0.56 | - |
| | Ours ($\alpha = 5e-6$) | **50.48** | **-0.23** | **-0.22** |
| | Ours ($\alpha = 2e-5$) | **54.58** | **0.17** | **0.04** |

## 4.4  Visualizing the pruned filters

We visualize the filters of VGG19 to show what the sparse network look like after pruning by SWP. The kernel size of VGG19 is $\mathbb{R}^{3\times3}$, thus there are 9 strips in each filter. Each filter has $2^9$ forms since each strip can be removed or preserved. We display the filters of each layer according to their frequency of each form. Figure 7 shows the visualization results. There are some interesting phenomenons:

- For each layer, most filters are directly pruned with all the strips.

- In the middle layers, most preserved filters only have one strip. However, in the layers that close to input, most preserved layers have multiple strips. Suggesting the redundancy most happens in the middle layers.

We believe this visualization may towards a better understanding of CNNs. In the past, we always regard filter as the smallest unit in CNN. However, from our experiment, the architecture of the filter itself is also important and can be learned by pruning. More visualization results can be found in the supplementary material.

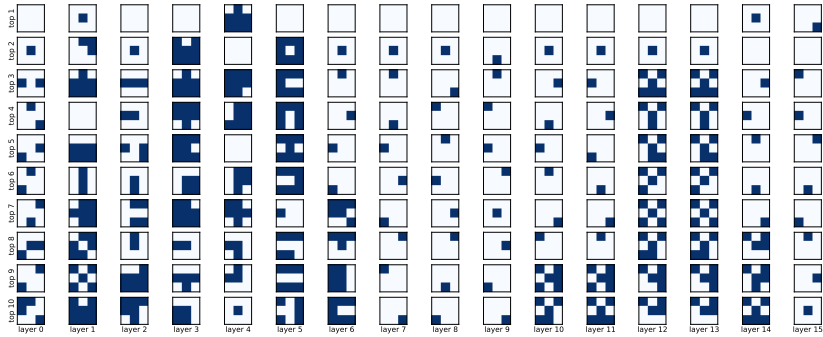

Figure 7: The visualization of the VGG19 filters pruned by SWP. From top to bottom, we display the filters according to their frequency in such layer. White color denotes the corresponding strip in the filter is removed by SWP.

Table 4: This table shows how $\alpha$ and $\delta$ affects SWP results. The experiment is conducted on CIFAR-10. The network is ResNet56.

| $\alpha$ | 0.8e-5 | 1.2e-5 | 1.4e-5 | 1e-5 | | | | |
|---|---|---|---|---|---|---|---|---|
| $\delta$ | | 0.05 | | 0.01 | 0.03 | 0.05 | 0.07 | 0.09 |
| Params (M) | 0.25 | 0.21 | 0.2 | 0.45 | 0.34 | 0.21 | 0.16 | 0.12 |
| Flops (M) | 61.16 | 47.71 | 41.23 | 111.68 | 74.83 | 56.10 | 41.59 | 29.72 |
| Accuracy(%) | 92.73 | 92.43 | 92.12 | 93.25 | 92.82 | 92.98 | 92.43 | 91.83 |

## 4.5 Ablation Study

In this section, we study how different hyper-parameters affect pruning results. We mainly study the weighting coefficient $\alpha$ in (1) and the pruning threshold $\delta$. Table 4 shows the experimental results. We find $\alpha = 1e-5$ and $\delta = 0.05$ gives the acceptable pruning ratio and test accuracy.

## 5 Conclusion

In this paper, we propose a new pruning paradigm called SWP. Instead of pruning the whole filter, SWP regards each filter as a combination of multiple stripes (*i.e.*, $1 \times 1$ filters), and performs pruning on the stripes. We also introduce Filter Skeleton (FS) to efficiently learn the optimal shape of the filters for pruning. Through extensive experiments and analyses, we demonstrate the effectiveness of the SWP framework. Future work can be done to develop a more efficient regularizer to further optimize DNNs.

**Acknowledgements:** This work was supported in part by the Guangdong Basic and Applied Basic Research Foundation under Grant 2019Bl515120055, in part by the Shenzhen Fundamental Research Fund under Grant JCYJ201803061720239​49, in part by the Open Project Fund AC01202005018 from Shenzhen Institute of Artificial Intelligence and Robotics for Society, and in part by the Medical Biometrics Perception and Analysis Engineering Laboratory, Shenzhen, China.

## Broader Impact

Advantage of our project:

- Training deep neural networks require a huge amount of time and resources. Pruning can reduce the network to a small size, thus resulting in reducing the cost of training. More importantly, our method does not need retraining (fine-tuning) which is more efficient in training deep neural networks.

We are not aware of a negative impact on our society. However, the pruned network may lose performance compared to an untouched state. Thus some hyper-parameters may need to be tuned very carefully.

## Footnotes

[1]In the author list, * denotes that authors contribute equally; † denotes corresponding authors. The work is conducted while Fanxu Meng works as an internship at Tencent Youtu Lab.

[1]https://github.com/pytorch/examples/tree/master/imagenet

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
