[Supplementary Material]

# Supplementary Material for 'Pruning Filter in Filter'

**Fanxu Meng**[1,2*]**, Hao Cheng**[2*]**,**
**Ke Li**[2]**, Huixiang Luo**[2]**, Xiaowei Guo**[2]**, Guangming Lu**[1†]**, Xing Sun**[2†]
[1] Harbin Institute of Technology, Shenzhen, China
[2] Tencent Youtu Lab, Shanghai, China
18S151514@stu.hit.edu.cn, luguangm@hit.edu.cn
{louischeng, tristanli,huixiangluo,scorpioguo,winfredsun}@tencent.com

## Abstract

This is the supplementary material for the paper 'Pruning Filter in Filter'. Section 1 shows that we can use SWP to continue pruning the network pruned by other FP (filter pruning) based methods. Section 2 shows that Filter Skeleton is better than lasso regularization in the SWP framework. Section 3 displays more visualization results.

## 1 Continual Pruning in SWP

Since SWP can achieve finer granularity than traditional filter pruning methods, we can use SWP to continue pruning the network pruned by other methods without obvious accuracy drop. Table 1 shows the experimental results. It can be observed that SWP can help other FP-based pruning towards higher pruning ratios.

Table 1: This table shows variant pruning methods on CIFAR-10 dataset. $A + B$ denotes that first prune the network with method A, then continue pruning the network with method B.

| Backbone | Metrics | FLOPS (M) | Params (M) | Accuracy |
|---|---|---|---|---|
| | Baseline | 14.72 | 627.36 | 93.63 |
| VGG16 | Network Slimming [1] | 1.44 | 272.83 | 93.60 |
| | **Network Slimming + SWP** | **1.09** | **204.02** | **93.62** |
| | Baseline | 20.04 | 797.61 | 93.92 |
| VGG19 | DCP[2] | 10.36 | 398.42 | 93.6 |
| | **DCP+SWP** | **3.40** | **253.24** | **93.4** |
| | Baseline | 0.86 | 251.49 | 93.1 |
| ResNet56 | GBN [3] | 0.30 | 112.77 | 92.89 |
| | **GBN+SWP** | **0.24** | **81.26** | **92.67** |

## 2 Filter Skeleton *v.s.* Group Lasso

In the paper, we use Filter Skeleton (FS) to learn the optimal shape of each filter and prune the unimportant stripes. However, there exist other techniques to regularize the network to make it sparse. For example, Lasso-based regularizer [4], which directly regularizes the network weights. We offer a comparison to Group Lasso regularizer in this section. Figure 1 shows the results. We can see under the same number of parameters or Flops, PFF with Filter Skeleton achieves a higher performance.

Figure 1: Comparing Filter Skeleton with Lasso regularizer on CIFAR-10. The backbone is VGG16.

Figure 2: In the figure, We exhibit the ratio of remaining stripes of each layer. Each filter has 9 stripes indexed from $s_1$ to $s_9$.

# 3 More Visualization Results

In this section, we show how the pruned network look like by SWP. Figure 2 shows the visualization results of ResNet56 on CIFAR-10. It can be observed that (1) SWP has a higher pruning ratio on the middle layers, *e.g.,* layer 2.3 to layer 2.9. (2) The pruning ratio of each stripe is different and varies on each layer. Table 2 shows the pruned network on ImageNet. For example, in layer1.1.conv2, there are original 64 filters whose size is $\mathbb{R}^{62\times3\times3}$. After pruning, there exists 300 stripes whose size is $\mathbb{R}^{62\times1\times1}$. The pruning ratio in this layer is $1 - \frac{300\times62\times1\times1}{64\times62\times3\times3} = 0.47$.

Table 2: This table shows the structure of pruned ResNet18 on ImageNet.

| keys | modules |
|---|---|
| (conv1): | Strip(3,324) |
| (bn1): | BatchNorm(64) |
| (layer1.0.conv1): | Strip(64,102) |
| (layer1.0.bn1): | BatchNorm(57) |
| (layer1.0.conv2): | Strip(57,164) |
| (layer1.0.bn2): | BatchNorm(64) |
| (layer1.1.conv1): | Strip(64,175) |
| (layer1.1.bn1): | BatchNorm(62) |
| (layer1.1.conv2): | Strip(62,300) |
| (layer1.1.bn2): | BatchNorm(64) |
| (layer2.0.conv1): | Strip(64,475,stride=2) |
| (layer2.0.bn1): | BatchNorm(119) |
| (layer2.0.conv2): | Strip(119,636) |
| (layer2.0.bn2): | BatchNorm(128) |
| (layer2.1.conv1): | Strip(128,662) |
| (layer2.1.bn1): | BatchNorm(128) |
| (layer2.1.conv2): | Strip(128,648) |
| (layer2.1.bn2): | BatchNorm(128) |
| (layer3.0.conv1): | Strip(128,995,stride=2) |
| (layer3.0.bn1): | BatchNorm(252) |
| (layer3.0.conv2): | Strip(252,1502) |
| (layer3.0.bn2): | BatchNorm(256) |
| (layer3.1.conv1): | Strip(256,1148) |
| (layer3.1.bn1): | BatchNorm(256) |
| (layer3.1.conv2): | Strip(256,944) |
| (layer3.1.bn2): | BatchNorm(256) |
| (layer4.0.conv1): | Strip(256,1304,stride=2) |
| (layer4.0.bn1): | BatchNorm(498) |
| (layer4.0.conv2): | Strip(498, 2448) |
| (layer4.0.bn2): | BatchNorm(512) |
| (layer4.1.conv1): | Strip(512, 3111) |
| (layer4.1.bn1): | BatchNorm(512) |
| (layer4.1.conv2): | Strip(512, 2927) |
| (layer4.1.bn2): | BatchNorm(512) |
| (fc): | Linear(512,1000) |

## Footnotes

[1]In the author list, * denotes that authors contribute equally; † denotes corresponding authors. The work is conducted while Fanxu Meng works as an internship at Tencent Youtu Lab.