[Reviews · NeurIPS 2020]

Review 1

Summary and Contributions: Different to the conventional channel pruning or shape-wise pruning, the paper proposed a new pruning method (PFF) that learns the optimal shape of each filter and performs stripe selection in each filter. An efficient implementation of the pruning method was also introduced.

Strengths: 1. The idea is novel. Different to the conventional channel pruning or shape-wise pruning, the paper proposed a new pruning method that learns the optimal shape of each filter and performs stripe selection in each filter. 2. The claims are reasonable that keeps each filter independent with each other which does not break the independent assumption among the filters. 3. Pruning stripes in filters can lead to regular sparse computation and can easily achieve acceleration in general processors.

Weaknesses: I have several questions: 1. Some SOTA pruning methods were not compared, such as [1]. 2. The actual inference latency of pruned networks is not provided. [1] Yu, Jiahui, and Thomas S. Huang. "Universally slimmable networks and improved training techniques." Proceedings of the IEEE International Conference on Computer Vision. 2019. [2] Chin, Ting-Wu, et al. "Towards Efficient Model Compression via Learned Global Ranking." Proceedings of the IEEE/CVF Conference on Computer Vision and Pattern Recognition. 2020.

Correctness: The claims and method are correct. The empirical methodology is correct.

Clarity: The paper is well written and easy to follow.

Relation to Prior Work: The paper has well discussed with related works including weight pruning, channel pruning and shape-wise pruning. Especially for shape-wise pruning [1] which removes the weights located in the same position among all the filters, the proposed method keeps each filter independent with each other, thus can lead to a more efficient network structure. The propsoed method utilizes an Filter Skeleton (FS) to prune spatial groups. FS can be viewed as a mask which determines whether preserve one location or not. The full-stack filters [2] which also utilize the mask to determine the states of weights in filters. The difference with [2] should be discussed. [1] Wei Wen, Chunpeng Wu, Yandan Wang, Yiran Chen, and Hai Li. Learning structured sparsity in deep neural networks. In Advances in neural information processing systems, pages 2074–2082, 2016. [2] Han, Kai, et al. "Full-stack filters to build minimum viable cnns." arXiv preprint arXiv:1908.02023 (2019).

Reproducibility: Yes

Additional Feedback: Update after rebuttal: I have read the rebuttal and other reviews. I keep my score as 6. The paper has novelty that learns optimal shape for each filter and provides efficient implementation. I also agree with other reviewers that there are still some presentation and claims unclear. I tend to vote for accepting this submission.


Review 2

Summary and Contributions: This paper proposes the pruning method PFF(pruning filter in filter) pruning stripes of filter instead of the whole filter using L1 regularization on filter skeleton. Here, filter skeleton is a soft mask of dimension n_l x K x K (for the number of filter n_l and kernel size K x K) which multiplies to weight of convolution filter and determines which stripes of length c_l (the number of channels in a filter) would be pruned. PFF trains the neural network with L1 regularization on filter skeleton, and prunes stripes by thresholding filter skeleton. PFF achieves finer granularity while being hardware friendly. Also, PFF achieves the sota pruning ratio on CIFAR-10 and ImageNet datasets without much accuracy drop.

Strengths: The paper is well-written and easy to follow. The main idea of the paper is simple and achieves good performance. Since implementation details are written in detail and attached code is well-explained, it is easy to follow the whole process of the paper end to end.

Weaknesses: First, the concept of pruning filter axes of convolution filter was introduced in [Kang, 2019]. So, PFF is not the first to prune filter axes of convolution filter. And, shape-wise pruning methods prunes filter axes of convolution filter and PFF prunes channel axes of convolution filter. L100-108 explains the benefit of pruning channel axes rather than filter axes but the experiments comparing these two kinds of method are very insufficient. Thus, substantially more experiments comparing shape-wise pruning methods and PFF required to make the case. Additionally, filter skeleton and soft mask used in [24] also shares the similar concept. In summary, the novelty of the proposed approach is very limited. ​ [Kang] Kang, H. J. (2019). Accelerator-aware pruning for convolutional neural networks. IEEE Transactions on Circuits and Systems for Video Technology. [13] Vadim Lebedev and Victor Lempitsky. Fast convnets using group-wise brain damage. In Proceedings of the IEEE Conference on Computer Vision and Pattern Recognition, pages 2554–2564, 2016. [24] Shaohui Lin, Rongrong Ji, Chenqian Yan, Baochang Zhang, Liujuan Cao, Qixiang Ye, Feiyue Huang, and David Doermann. Towards optimal structured cnn pruning via generative adversarial learning. In Proceedings of the IEEE Conference on Computer Vision and Pattern Recognition, pages 2790–2799, 2019.

Correctness: The claims and methods in the paper are correct. The paper prunes model with same baseline model used in baseline methods and evaluates pruned model with usual image classification tasks.

Clarity: The paper is well-written.

Relation to Prior Work: The paper sufficiently covers the prior works and discusses a relationship between them.

Reproducibility: Yes

Additional Feedback: As I mentioned before, in order to claim pruning filter axes is better than pruning channel axes, the substantial experiments comparing shape-wise pruning methods and PFF are required to make the case. ----- Thank you for your answers to the feedback. I still don't think that the stripe-wise pruning method is new, but I agree that analyses in this paper and the performance of the method are worth being heard about in the efficient inference community.


Review 3

Summary and Contributions: The paper proposes a method to "prune the filter in the filter (PFF)" that prunes a trained neural network. It treats a tensorial filter as multiple 1x1 filters, and learns to remove the 1x1 filters through a sparsity-constrained optimization. It additionally introduces "Filter Skeleton" wholse values reflect the optimal shape of each filter. It demonstrates the proposed PFF achieves state-of-the-art pruning ratio on CIFAR10 and ImageNet datasets without accuracy drop.

Strengths: The idea of decomposing tensorial filters into "Filter Skeleton" and sparse factors is interesting. It resembles tensor decomposition but exploits a discriminative objective function Eq. (1). It makes sense to prune the decomposed 1x1 filters while maintaining a fairly reasonable "Filter Skeleton", which can be thought of as high-order PCA on tensors.

Weaknesses: The paper introduces several new concepts without clarifying well, such as "Filter Skeleton" and "optimal shape". This makes the paper a little hard to follow. Here are some other questions that need to answer or clarify. "Fine-tuning" seems like an important step in standard filter pruning pipeline as stated in Line31. The authors claim that PFF keeps "high performance even without a fine-tuning process" in Line152? But the paper does not compare the performance with and without fine-tuning. So how the conclusion is derived? Equation (1) says that the filter pruning requires a "training" step over the train set, then why it is advantageous to skip fine-tuning network parameters, given that solving Eq. (1) still requires backpropagating through the whole network? Eq.1 and Figure 2: It is not clear why using dot product? Is W*I reconstructing the original filter? It is not clear what "shape" means in Line10, Line42 and Line200. Can the authors clarify? Perhaps the concept is known for specific audience, it is not self-explanatory in this paper. In Line200, it is still not clear what the "shape of the filter" means and why the shape could matter? Should it be expected to matter or not? Why does it show "shape" matters from Figure 4? By line203, it seems to indicate the performance is not affected by "shape"? Line178: It is not clear what "one-shot finetuning" means. Figure 5: What are the x-axes? Line209: How to tell "the weights of the network become more stable"? Figure 6: How to define "top-1", "top-2", ..., "top-10"? typo: Line154: "a optimal" --> "an optimal" Line155: "the filter do not lose much...." --> "the filter does not loss much..." Line 161: "how PFF prune the network" --> "how PFF prunes the network" Line172: "... implementation that train the model..." --> "... implementation that trains the model..." Line222: "may towards a better understanding"

Correctness: One claim in the paper is that PFF keeps "high performance even without a fine-tuning process" in Line152. But the paper does not compare the performance with and without fine-tuning. So it is unclear how the conclusion is derived. Moreover, Equation (1) says that the filter pruning requires a "training" step over the train set, then it is not clear why it is advantageous to skip fine-tuning network parameters, given that solving Eq. (1) still requires backpropagating through the whole network.

Clarity: The paper is a little hard to follow. It introduces several new concepts which are not clarified well. Figures 1 and 2 are hard to understand.

Relation to Prior Work: The paper presents related work fairly well.

Reproducibility: No

Additional Feedback: Addressing points listed in "Weaknesses" will improve the paper, w.r.t presentation and clarity. ----------------------------after rebuttal Thank authors for the rebuttal! I've read all the reviews and rebuttal. I'd maintain my initial rating. My concerns are still on the unclear presentation and claims in this paper. For example -- The authors answer my question(s) by pointing to the paper L112-L119 (but I do think they are referring to Line202-206): "We surprisingly find that the network still achieves 80.58% test accuracy with only 12.64% parameters left. This observation shows that even though the weights of filters are randomly initialized, the network still has good representation capability if we could find an optimal shape of the filters. After learning the shape of each filter, we fix the architecture of the network and finetune the weights. The network ultimately achieves 91.93% accuracy on the test set." 1. It reads like that fine-tuning is still crucial (authors mention CIFAR10 in Figure 4 caption when writing the paragraph including the above sentences). 2. It is not clear what "randomly initialized" means. If randomly initialized, why not fine-tuning as re-iterated by the authors? 3. When authors report 80.58% accuracy with only 12.64% parameters left and claim this could be an "optimal shape", I would like to see performance by "suboptimal shapes" with similar portion of parameters left (e.g., randomly dropping filters). This can help understand how/why "optimal shape" matters or not. In other word, the paper does not provide with a comparison to justify this "optimal shape" claim.

[Author Response · NeurIPS 2020]

**Response to all reviewers:** We thank all reviewers for their valuable and thoughtful comments. R #1 and R #4 describe
our work as novel and interesting while requiring further concept explanation and performance comparison. R #2
appreciates the writing part but has concerns about the novelty. Below, we first clarify the novelty of this paper in
response to the potential misunderstanding, then we present detailed answers for each reviewer.

Our main contributions are twofold: 1) We propose a new pruning paradigm called pruning filter in filter (PFF), which
is a stripe-wise pruning and can be seen as a general case of the filter-pruning (FP). PFF treats a filter $F \in \mathbb{R}^{C \times K \times K}$ as
$K \times K$ stripes, *i.e.*, $1 \times 1$ filters $\in \mathbb{R}^C$ and perform pruning at the unit of stripe instead of the whole filter. Compared to
the existing methods, *i.e*, filter pruning (FP) and weight pruning (WP), PFF achieves finer granularity than traditional
FP while being hardware friendly than WP, leading to state-of-the-art pruning ratio on CIFAR-10 and ImageNet. 2)
More arousingly, by applying PFF, we find another important property of filters alongside their weight: *shape*. Start
with a random initialized ResNet56, we train and trim the shape of filters and boost the test accuracy from 10.00% to
80.58% on CIFAR-10 without updating the filter weights. The optimal shape of filters are learned by the proposed Filter
Skeleton (FS) in the paper and we believe FS could inspire further research towards the essence of network pruning.

**Response to Reviewer #1: Q1:** The paper does not compare to [1]. **A1:** [1] only uses MobileNet. However, we follow
most pruning papers and use ResNet and VGG to perform experiments in the paper. Due to limited time during rebuttal,
we only perform pruning with MobileNetV2 on ImageNet under PFF setting. PFF achieves 70.2% accuracy with 160M
Flops, while [1] reported 67.1% accuracy with 167M Flops. We will conduct more experiments in the revised version.
**Q2:** Inference time is not provided. **A2:** Since we modify the calculation order in convolution, only Python-version
code (in the attached supplementary material) without re-implementing CUDA code can not reflect the actual inference
time for PFF. In the future, we will re-implement CUDA code to provide the actual inference time of PFF. **Q3:** The
difference between filter skeleton and [2]. **A3:** 1) [2] uses the binary mask to convert the full-stack filter to multiple
sub-filters and one filter corresponds to multiple masks. However, FS learn the optimal shape of each filter and one
filter corresponds to one mask; 2) The value of maks in [2] takes value from {-1,1}, thus all the weights of the filter are
used. While the values of FS are continuous and are made sparse during training. The stripes corresponding to 0 in FS
will be pruned; 3) The assumption in [2] is that the filters should be mutually orthogonal. While the assumption in FS is
that the filters should be independent with each other. We will cite [1,2] and add these analyses to the paper properly.

**Response to Reviewer #2: Q1:** Is PFF the first to prune channel axes of convolution filter ?**A1:**[Kang, 2019,
Accelerator-aware pruning for convolutional neural networks] does mention the concept of pruning channel axes
(stripe) of convolution filters. However, they divide each stripe into fetching groups and perform weight pruning
within each group, which is a hardware implementation optimization for weight pruning. To our best knowledge,
PFF is the first stripe-wise pruning method. PFF only prunes the weight along channel axes (filter strip) and achieves
state-of-the-art pruning ratio while being hardware friendly. We will cite this work and add these analyses to the paper
properly. **Q2:** The experiments between shape-wise pruning and PFF are insufficient. **A2:** Besides the experiments
in Table 1, we also perform experiments on shape-wise pruning and PFF in the supplementary material. In Figure 2
of the supplementary material, stripe-wise pruning (PFF) and shape-wise pruning are conducted in exactly the same
setting. We can see that under the same number of parameters or Flops, PFF achieves a higher performance compared
to shape-wise pruning. **Q3:** The difference between FS and soft mask in [24]. **A3:** Filter Skeleton (FS) in PFF is not
just a soft mask. The mask in [24] is to learn the importance of each channel. While we consider each filter has two
properties: weight and shape. FS is to learn the 'shape' property. From Section 4.3 in the paper, the network still has
good performance by only learning the 'shape' of the filters, keeping the filter weight randomly initialized.

**Response to Reviewer #4: Q1:** what does the shape and optimal shape mean? **A1:** For a filter $F \in \mathbb{R}^{C \times K \times K}$. There
are $K \times K$ stripes. Each stripe can be kept or removed. Thus there are $2^{K^2}$ shapes of this filter. The optimal shape of the
filter is the filter with minimal stripes that keeps maximal useful information. **Q2:** Why the optimal shape matters? **A2:**
In L112 → L119, network with an optimal shape and 12.64% random initialized weight still achieves high performance,
which indicates that a good shape of filters can still learn useful knowledge from data. Thus the optimal shape matters
in pruning. **Q3:** Why use dot product in Eq (1) in the paper? **A3:** Filters used for extracting features are produced by
multiplying $W$ and $I$ (dot product) in Eq (1). Similar notation is used in *https://arxiv.org/abs/1908.02023*. **Q4:** Why
PFF does not need fine-tuning? **A4:** Weights of each stripe and FS are multiplied (dot product) and jointly optimized.
Thus the stripes corresponding to 0 value in FS have no contribution to the output, which can be removed accordingly.
We find that the fine-tuned network and the network without fine-tuning end up with very similar results on CIFAR-10,
indicating the effectiveness of the learned filter shape. Thus as mentioned in L177, a post fine-tuning is not necessary
on CIFAR-10. **Q5:** what is the x-axes in Figure 5? Why the weights trained by PFF is more stable? **A5:** The x-axes is
the number of filters. It can be seen that the weights trained by PFF are sparse and smooth, which have a low variance
to input images, leading to stable outputs. **Q6:** what does one-shot fine-tuning means? **A6:** One-shot means that we
only train the pruned network once until convergence. **Q7:** How to define top-1 in Figure 6? **A7:** Top-1 indicates the
highest shape frequency in such layer. We will make all the presentations more clear in the revised version.

[Meta-Review · NeurIPS 2020]

The paper presents a simple method with good empirical results. The authors have done a thorough job of investigating what goes on in the method. This is all good and acceptance is therefore recommended. However, the reviewers are not happy with the presentation and some of the claims in the paper. The authors for their own sake need to address this for the poster and the camera ready version of the paper.